# A Rapid Review of Sociocultural Dimensions in Nigeria’s Solid Waste Management Approach

**DOI:** 10.3390/ijerph20136245

**Published:** 2023-06-28

**Authors:** Thomas Akintayo, Juha Hämäläinen, Pertti Pasanen, Iniobong John

**Affiliations:** 1Department of Environmental and Biological Sciences, Faculty of Science, Forestry, and Technology, University of Eastern Finland, Yliopistonrantia 1, FI-70210 Kuopio, Finland; pertti.pasanen@uef.fi; 2Department of Social Sciences, Faculty of Social Sciences and Business Studies, University of Eastern Finland, Yliopistonrantia 1, FI-70210 Kuopio, Finland; juha.hamalainen@uef.fi; 3Department of Quantity Surveying, Faculty of Environmental Sciences, University of Lagos, Lagos 100213, Nigeria; ijohn@unilag.edu.ng; 4SARChl in Sustainable Construction Management and Leadership in the Built Environment, Faculty of Engineering and the Built Environment, University of Johannesburg, Johannesburg 2006, South Africa

**Keywords:** circular economy, interdisciplinarity, recycling, refuse management, sociocultural intervention, sustainability

## Abstract

Against the background of an arguable dearth of scholarship on the sociocultural dimensions of Nigeria’s solid waste management strategies and practices, this rapid review searched for evidence in the literature. A rapid evidence review and qualitative meta-summary procedure were implemented to utilize the rigor of systematic literature review that met the timelines and limited funding available for this study. It is more appropriate to identify, extract, and synthesize a mixture of qualitative and/or quantitative empirical evidence in the literature. This rapid review found little substantive evidence of scholarly sociocultural approaches in Nigeria’s solid waste management. It also discovered constant factors of inadequate and weak multidisciplinary or non-holistic approaches to driving innovation and effective social impact in Nigeria’s solid waste management practices. The results were interpreted vis-à-vis the need to leverage the social sciences, particularly the range and scope of social work practice configurations and possibilities, to scientifically advance and substantially accelerate the implementation of evidence-based policy and practice in Nigeria’s solid waste management system. This rapid review concluded that the negative results are due to the insufficient conceptual and theoretical bases for Nigeria’s solid waste management strategies and/or practices.

## 1. Introduction

Arguably, there is a dearth of scholarship on the sociocultural dimensions of Nigeria’s solid waste management (SWM) strategies. Whereas, to reduce waste, interventions that draw on the social sciences are needed, particularly science of behavior change [1], which may be realized, for example, by community sustainability pledges and training, civic orientation, children literature, green movements, and programs in schools, as in many developed countries. Citizens’ awareness of their roles and responsibilities in Nigeria’s SWM system is necessary for positive outcomes. However, their roles and responsibilities seem not often the inclusive focus of scholarly work, resulting in persistent waste disposal challenges.

In other words, (solid) waste management, in particular, generally attracts multiple disciplines because it is a major area of public policy and a subject of a burgeoning area of multidisciplinary academic research [2]. The social sciences, particularly civic education or citizens’ orientation and behavioral tendency toward recycling, reusing, upcycling, and reducing household or municipal and industrial solid wastes play major roles, but they are seemingly not inclusively prominent in scholarly approaches to SWM in many (developing) countries, including Nigeria. Therefore, there is a need for a rapid evidence review (RER) of likely interdisciplinarity and/or transdisciplinary and multidisciplinary approaches in the literature on Nigeria’s SWM system to ascertain the level of involvement of multiple disciplines, particularly the social sciences in Nigeria’s SWM approaches. The RER should include a search for the magnitude of the evidence, or the frequency effect size (FES) [3], and the impact of each study, or the intensity effect size (IES) [4], of possible sociocultural factors in the literature on Nigeria’s SWM practices. Evidence of focus on advocacy, individual and/or group inclination and participation, and their community values, beliefs, orientation, knowledge, points of view, and related modalities and conditionalities, etc., that is, the inclusion of social sciences/social work perspectives in Nigeria’s SWM approaches, should be included in the search. Thus, the type and scope of data needed and the research funder’s time constraints warranted the study’s rapid review method [5,6] and qualitative meta-summary (see [7,8]). The two techniques synthesized the best available descriptive evidence in electronically indexed literature produced via collaborations, particularly highlighting the scholarship impact of likely sociocultural factors in Nigeria’s SWM practices.

### 1.1. Background

The economic and political foundations of societies have, over time, distorted nature. Even modern societies often intentionally and accidentally violate urban and suburban architecture and/or planning and designs that are put in place to mitigate human distortion of nature. Thus, we have huge behavioral (and subsequently, some cultural) problems that are inimical to the sustainability of contemporary societies. In particular, waste disposal (either indiscriminate or accidental) across the world—including Nigeria’s solid waste (among other pertinent variables, such as loss of biodiversity, natural resource depletion, air pollution, population growth and movement, urbanization, and poverty)—is a categorical part of the global environmental sustainability challenge. (See page 13 of the Scientific Committee on Problems of the Environment (SCOPE) in the United Nations Environment Programme [9] for a full list of 21st-century emerging issues as survey variables.) Akan et al. [10] posited that only between 9% and 12% of the total generated waste in Nigeria is recycled or incinerated, which is inadequate. Moreover, in a recent study, existing SWM practices in Global South cities resulted in land degradation, air, and water pollution, climate change, methane emissions, and hazardous leachate [11]. Thus, solid waste management should focus not only on environmental protection but also on the sustainability of contemporary social and economic environments and general well-being.

The environmental problem of municipal solid waste in Nigeria, consisting of paper, metal, glass, textile, plastic, ash dust, organic content, and other micro-elements [12], is estimated at 0.65–0.95 kg/capita/day [13] or 66,828 tons/day for a total urban population of 106 million [14]. This is a substantive challenge threatening the sustainability or Sustainable Development (SD) of Nigeria—Africa’s most populous nation, which is responsible for more than half of the 62 million tons/year of waste generated in sub-Saharan Africa. Interestingly, contemporary deliberations on SWM have been replaced with discussions on circular economy (CE) principles—best epitomized in the idea that nature does not waste anything and everything circulates [15]—and which are supported by the global adoption of the concept of SD [16]. Coincidentally, one of the advocated strategies for SD is interdisciplinary research [17].

In particular, a multiple disciplinary approach to research seems to have created the basis for a systems-thinking perspective in SWM practices generally and for knowledge and innovation in SD locally. Both seem applicable to waste management challenges in Nigeria. Although the concept of sustainability has been postulated to have multiple meanings and to be in danger of overuse [18] (see also [19]), it is seemingly untrue if the relationship among the main concepts of SD is examined. Since 1987, when SD was emphasized in the Brundtland Report as the enhancement of economic growth and a possible solution to the negative effects of industrialization and population growth (see the Report of the World Commission on Environment and Development [20]), concepts related to SD have been developed. Currently, three main related concepts—environmental sustainability, economic sustainability, and social sustainability—coexist, as illustrated in Figure 1. 

The interdisciplinarity and hierarchy of relationships among the SD concepts, shown in Figure 1, may have engineered the evolution of the Sustainable Development Goals (SDGs) of the United Nations’ 2030 Agenda. Recently, the hierarchy in relationships prompted [21] (p. 3) to postulate that “SD itself is the principle for attaining sustainability”. In other words, SD postulated in the Brundtland Report as the enhancement of economic growth and mitigation of the negative effects of industrialization and population growth (which in modern dispensations are sustainable economy, quality of life, conserving the environment/landscape, air, seascape (i.e., seas and oceans), etc.) has become the foundational idea or a rule that controls the attainment of sustainability. Figure 1 also shows that two academic departments could be essential for SD—the environmental sciences and the social sciences (the social and economic/CE as an example of the multidisciplinary approach in an SD intervention)—and omit its real infrastructural and technological aspects. However, the scope is beyond the two departments, thus the need for a multiple disciplinary or systems-thinking approach to SD, and by extension to Nigeria’s SWM services, and to elsewhere globally.

Subsequently, of concern in this research is the possibility for developing countries, particularly Nigeria’s SWM system, to leverage the idea (pertinent to the CE) that nature does not waste anything and more of the relevant socioeconomic transitions obvious in many developed and industrial economies. Some interventions in industrial economies from linear to circular are to reduce waste or dwindling economic and ecological resources and to control the negative effects of increasing global warming and environmental stressful situations, pollution resulting from mining accidents, and poor WM practices. Such a transition embraces industry, innovation, and infrastructure (SDG9) in line with [22] SDG Target 9.A and possibly Responsible Consumption and Production (SDG12). In this regard, sustainability transition models (see [23] for an example) with implications for SWM systems have been developed. In particular, there is the multi-level perspective (MLP) sociotechnical system, which “involves interactions between three groups: niche innovations, sociotechnical regime, and sociotechnical landscape” [24] (p. 3). Generally, the sociotechnical transition has been defined as a set of fundamental change processes that have technological, cultural, consumer practices, markets, supply chains, regulations, and infrastructures, among others, which is a clear example of interdisciplinarity or transdisciplinary and multidisciplinary approaches to SD vis-à-vis global ecological challenges. In summary, the increasing global trend of providing multidisciplinary solutions (see [25]) to global warming and dwindling ecological resources is a reality. However, the environmental sciences still seem to dominate SD issues.

In contrast, recently some scholars of social work—which is a social scientific discipline in continental Europe—have over time established a conceptual relationship between eco-social sustainability and social work, as well as the role the profession is playing in promoting the issue of sustainability. However, there is no evidence that such an approach has been leveraged in Nigeria’s SWM approaches. For example, Rambaree et al. [26] explored how social work can change from an anthropocentric paradigm to an eco-social work paradigm to promote social change in community practice. Ramsay and Boddy [27] identified and explained the key attributes of environmental social work to emphasize that the adaptation of social work methods to promote social change can help create and sustain a biodiverse planetary ecosystem. Thus, social work’s analysis and intervention in the interface between the human and physical environments have been conceptualized via different terminologies such as “Green Social Work” in [28,29,30], among others, and “Environmental Social Work”, as in [27,31,32], to mention a few. Other conceptual terminologies include “Sustainable Social Work,” as in [33,34,35], and “Ecological Social Work” in [36,37,38]. All the terminologies are basically concerned with the impact of human activity on its physical environment, one of the pillars of SD. In addition, SD is also essentially an ethical and legal issue, not only in terms of political and professional expertise, and possibly in agreement with [39] (p. 8) that “environmental justice as a social work issue”—the deprivation of people of their environmental rights—is an example of the expanded reach of social policy.

In other words, environmental issues are inclusively an issue of citizens’ participation and patriotism, visionary leadership, and honesty, which are of concern to professional social work. It leverages the social science perspectives or focuses on attitudes, beliefs, cultural values, knowledge, and many more to foster social development and/or the well-being of individuals, groups, and communities that may be experiencing sustainability or environmental challenges, which is contrary to the view of [18] (p. 20), who posited an ironic question: “What has sustainability got to do with social work?” Of course, their own answers to the question were used to elaborate on the indispensability of social work practice to leverage social sustainability as a means for long-term global security. Therefore, environmental science perspectives alone can no longer be dominant in the promotion of the global sustainability agenda or SDGs. Furthermore, the social work profession’s concern with SD is just one of how the issues of SD pose new challenges and tasks for traditional professions. A similar conceptualization or multidisciplinary approach and the development of a new professional paradigm based on the philosophy of SD seem widespread in other fields as well. However, the social work profession in Nigeria (and in many developing countries) has been struggling to exist legally (see [21]), and it was not until the last quarter of 2022 that the profession was backed by Acts of Parliament. Thus, considering the profession’s potentials for promoting social change to sustain the biodiverse planetary ecosystem, not much is known of the profession’s role and the level of involvement of other social sciences perspectives in SWM practices in Nigeria, particularly their interventional modalities and conditionalities. Therefore, there is a need for a rapid review.

### 1.2. The RER Conceptual Framework and Definition of Sociocultural Dimensions

The issues of SD, or especially SWM services, appear at the intersection of architectural design and planning of the environment, policies, and laws, economics, technology, and transitions or innovation as well as the interactions of stakeholders. In particular, because Nigeria’s SWM practices ought to be all-inclusive issues of many disciplines, the popular multi-level perspective on sociotechnical systems (e.g., see [23,24]) developed based on historical research and used as a strategic guide in systemic change process and positioning of relevant research services serves as this RER’s theoretical illustration of Nigeria’s solid waste practice landscape. The complex journey of waste—such as bio-waste, textiles, paper, plastic, glass, metals, carton, electronic, and mixed wastes, among others—which consumers may have to recycle (and possibly to grow the economy) and save natural resources can be quite complex. Thus, it has been argued that sociotechnical systems in waste management systems include not only technological dimensions but also far-reaching dimensions, such as cultural dimensions, consumer practices, markets, supply chains, regulations, and infrastructural dimensions, among others (see [23,40,41]). See the illustration in Figure 2.

In Figure 2, sociocultural factors are dominant across the three levels, which [42] characterized as follows: landscape (worldviews, paradigms, culture, and politics), *regime* (interacting institutional processes), and niche (domain of novelties). Therefore, the sociocultural dimension in this RER is defined as the influence of cultural and demographic characteristics, which include education, knowledge, religion, beliefs, laws, advocacy, governments, demographics, social classes, gender, lifestyles, behavior, and attitudes as well as individual, group, and community values and the indigenous approach to SWM practices in Nigeria. Consequently, the study includes a search for evidence of interdisciplinarity, transdisciplinary, and multidisciplinary approaches and sociocultural dimensions in Nigeria’s SWM system. Therefore, this study implemented an RER of qualitative and quantitative academic literature published between 1986 and 2022 generated from Scopus, Business Source Elite (EBSCO), EBSCOhost Academic Search Premier, and SocINDEX databases via UEF Primo (a search engine for printed and electronic materials at the University of Eastern Finland Library) for evidence of multiple disciplines and sociocultural focus on Nigeria’s SWM practices. The following research questions guided the RER:Q1: What evidence of interdisciplinarity and/or transdisciplinary and multidisciplinary approaches can be found in the literature on Nigeria’s SWM system?Q2: What inclusive evidence of sociocultural dimensions or social work/social science perspectives can be found in the literature on Nigeria’s SWM practices?Q3: What inclusive interventional modalities and conditionalities of sociocultural dimensions or social science perspectives can be found in the literature on Nigeria’s SWM strategies?

In other words, quantitative and qualitative peer-reviewed studies focusing on Nigeria’s SWM system could provide evidence of interdisciplinarity, or transdisciplinary and multidisciplinary approaches, and sociocultural factors or social science perspectives and their possible interventional modalities and conditionalities for this RER.

### 1.3. Significance of This Study

One contribution of this RER is the provision of a basis for seemingly needed evidence-based recommendations for sociocultural dimensions and sociotechnical transformation of Nigeria’s SWM strategies amid recurrent challenges of solid management in the country. Such a transformation could be replicable in other countries that need to strengthen their SWM systems, particularly developing countries. In addition, the potential outcomes of this study may strengthen the existing bases for the implementation of SDG9 (Industry, Innovation, and Infrastructure) and SDG12 (Responsible Consumption and Production) in Nigeria and elsewhere. Another significance of this RER is the focus on an advanced search for the literature on interdisciplinarity and/or transdisciplinary and multidisciplinary approaches to SWM vis-à-vis sociotechnical theory as a result of the need to systematically identify and highlight the main gaps and recurrent challenges in Nigeria’s SWM approaches amid the global trend for collaborating on issues concerning SD. Moreover, a simple search of indexed databases revealed a seeming dearth of standard literature reviews focused on multiple disciplines and sociocultural approaches to Nigeria’s SWM practices, either as systematic literature reviews (SLRs), quick scoping reviews (QSRs), evidence reviews (ERs), rapid qualitative reviews (RQRs), or rapid evidence assessments (REAs), which are possible options (see [3]). Finally, this study is a steppingstone for other new (social scientific) studies that require a broad focus on sociotechnical design for SWM in the framework of Finland-Lagos Recycling Culture Research & Development Project’s (F-LRCRDP) research series.

## 2. Methods

An overview of the major techniques implemented in this RER is as follows. First, the research questions were subjected to rapid review criteria. Second, an information specialist at the University of Eastern Finland’s (UEF) library was engaged, and search strategies were developed. Third, the Preferred Reporting Items for Systematic Reviews and Meta-Analysis (PRISMA) 2020 flow diagram often used in SLRs was deployed to identify and screen the literature generated from the indexed databases search. Thereafter, for generic and comparative typification, extraction, and analysis of evidence relevant to this RER, the desired characteristics in the literature reviewed were coded A, AB, AC, or ABC and ABCD (or in small letters abcd to indicate their possible variants) and vice versa in the manner of Boolean algorithms. See [43,44] for a full illustration and detailed logic, respectively, of the Boolean algorithms. Fourth, limited descriptive texts and statistics compatible with the qualitative meta-summary procedure were used in the reporting of the RER results.

### 2.1. Refining the Research Questions

To ensure that the research questions were well-defined and developed for focus and feasibility, they were subjected to the Feasibility, Impactful, Novel, Ethical, and Relevant (FINER) criteria developed by [45] as a necessary parameter in a rapid review or RER (see also [46]). The research questions fit the criteria and comprise six sub-categories—collaboration vis-à-vis interdisciplinarity, transdisciplinary and multidisciplinary approaches, sociocultural dimension or social science perspectives, and their interventional modalities and conditionalities.

### 2.2. Search Strategy, Data Sources, Search Results, and Selection Criteria

An information specialist at the UEF’s library conducted a search in electronic databases for indexed relevant journal publications. From 21 February 2022 to 29 September 2022, two search strategies were implemented. The first search implemented via UEF Primo was (“waste manag*” OR “municipal waste*”) AND TITLE (nigeria*). The search was limited to peer-reviewed literature, and 149 publications published between 1986 and 2022 were identified in the Scopus database. The second search was conducted on 29 September 2022 in the same search engine and included (circular econom* OR circular ecosystem* OR recycling) AND nigeria*. However, this search was limited by date (2010–2022) and academic journals, and it identified 76 publications in three databases: the Academic Search Premier, Business Source Elite, and SocINDEX. In other words, four electronic databases were searched via UEF Primo.

Subsequently, the characteristics of all identified publications—title, abstract, date, and authors’ information—were generated from quantitative and qualitative peer-reviewed literature (and some gray literature). The information was compiled in a Microsoft Word file and hard copies made for selection and screening. The following were the eligibility criteria for screening literature for the RER: peer-reviewed literature on Nigeria’s SWM system, practices, and approaches; peer-reviewed literature on the circular economy; and peer-reviewed literature published between 1 March 1986 and 29 September 2022.

In other words, any publication that did not exhibit the eligibility criteria above, such as gray literature, conference proceedings, or book chapters, was excluded. The PRISMA 2020 flow diagram, which is recommended for SLR processes and steps [47], was adapted to report the identification and screening of the publications retrieved for the RER, as illustrated in Figure 3.

### 2.3. Procedure for Analysis and Qualitative Meta-Summary of the Descriptive Evidence

Each work included in the rapid review (*n* = 70) was defined and was keyworded as follows: one, collaborative studies (two authors minimum) and coded A; two, trans-, inter-, or multidisciplinary approaches (two different disciplines or departments or faculties as minimum) and coded B; three, with evidence of sociocultural dimensions or social science perspectives and coded C, and four, with (their) interventional modalities and conditionalities and coded D.

The defining and coding of the characteristics made it easy to further group the literature to properly represent appropriate data. Thus, a four-item template comprising the title, author information, abstract, date, and primary paper was created to extract the data. For the extraction, typification, and evidence synthesis, a qualitative meta-summary procedure as suggested by [6] seemed appropriate to identify, extract, and synthesize a mixture of qualitative and/or quantitative empirical evidence and by extension the literature on multiple disciplinary and sociocultural dimensions in Nigeria’s SWM practices. In particular, the qualitative meta-summary created an opportunity for extraction and grouping of evidence and, possibly, the quantitative computation of the FES and the IES that mostly contributed to this review. In summary, the outcomes of the steps are shown in figures and tables, as presented in Section 3.

## 3. Results

The results for the grouping and abstracting of data are reported in this section, and a complete list of the literature included in the RER (*n* = 70) is in the Appendix A section (Table A1). In addition, while abstracting and grouping of evidence are displayed in Table 1, the FESs and IESs are displayed in Table 2. Detailed explanations of Table 1 and Table 2 vis-à-vis Figure 4 and Figure 5 are included in the following subsections.

### 3.1. Grouping and Abstracting the Evidence

First, the defining and coding of the included literature (*n* = 70) created two groups of relevant literature: Group 1, with ABCD characteristics (*n* = 52), and Group 2, with b (*n* = 12) indicating single authors from different disciplines but no interdisciplinarity, and those with BCD characteristics (*n* = 6) that provided relevant evidence, as indicated in Figure 4 and Figure 5, respectively.

In other words, the abstracting of evidence, referencing, and taking note of their frequencies focused on the relatively small number of studies, ABCD (*n* = 6) and ABC (*n* = 13), that are evidence of sociocultural dimension or social science perspectives, as well as the single-author papers (*n* = 6) of the BCD category. These patterns are more obvious in the illustration of the abstracting of evidence in Table 1 and in the illustration of the FESs and IESs in Table 2.

Furthermore, the evidence (or AE) in Table 1 is subjected to FES and IES computation to ascertain the relative magnitude of the abstracted evidence and the impact of each study in the reviewed literature. Thus, the outcomes of both steps are explained and illustrated in the following section.

### 3.2. Frequency and Intensity of the Effect Sizes

The results for the computation of how often particular evidence appeared in the reviewed literature or the relative magnitude of an AE, which is equivalent to the FES, are illustrated in Table 1. Similarly, the results for the computation for ascertaining the impact of each study—the IES of each study—in the reviewed literature is also illustrated in Table 2.

The outcomes of the methods implemented to ascertain and synthesize the best available descriptive evidence on probable sociocultural factors in the country’s SWM practices are further interpreted in Section 4.

## 4. Discussion

### 4.1. Meaning of the Findings

The results’ interpretations have different facets. First, the purpose of the rapid review was to analyze and synthesize the best available descriptive evidence from the literature, particularly of probable sociocultural factors, among others, in Nigeria’s SWM strategies. Almost all of the seventy works included (spanning about three decades) acknowledged the increasing SWM challenges in Nigeria, thus the need for a holistic strategy that combines the expertise of different disciplines, professions, government, and non-government intervention at different levels to devise an innovative solution that is line with current trends and the goals of SDGs of the United Nations’ 2030 Agenda.

Second, although there is substantive evidence of sociocultural dimensions in Nigeria’s SWM practices in the literature, the number of studies from 1986 to 2022 is too small. However, those few studies agree (1) that social science approaches are needed to effectively tackle the challenges of managing refuse. Those small number of studies also agree (2) that refuse management is a burgeoning area of interdisciplinarity and/or transdisciplinary and multidisciplinary academic research. The Scopus, Academic Search Premier, Business Source Elite, and SocINDEX databases together generated only six studies (excluding the BCDs) of the ABCD category, which is the best available evidence of probable sociocultural dimensions in Nigeria’s SWM approaches. In other words, the data presented in Section 3 indicate that there are a few studies with the ideal category of social–cultural dimensions in Nigeria’s SWM strategies. Specifically, the constant factors in the findings and/or recommendations of the very few studies are the inadequate and weak multidisciplinary or non-holistic approaches to driving innovation and effective social impact in Nigeria’s WM practices.

Third, challenges implementing research findings exist in Nigeria because SP9 (2020), SP16 (2018), SP40 (2002), SP46 (2019, SP47 (2018), and SP48 (2012)—all six in the ABCD category that represent the finest level of evidence for this review—were published over two decades (2002 to 2020). In addition, this connotation suggests that there is a huge gap between research and evidence-based policy practice in Nigeria. In contrast, in agreement with [26,27], the scale or range and scope of social work practice have many configurations and possibilities [43], which may be one great or main means, among others, for addressing the problems regarding SWM and individuals, groups, communities, businesses, and other social institutions that generate waste are concerned.

Therefore, as a matter of urgency, this paper recommends a strong multidisciplinary research and practice approach with strong cultural or social scientific input to lead the social impact team. In this case, the inherent expanded mandate of the social work profession for intervention in the physical environment agrees with the increasing multidisciplinary solution [25] to global warming and dwindling ecological resources worldwide. For the social work profession to fulfill its individual, environmental, and social change mandate [48], it draws theories from human development and behavior and social systems [49]. In other words, the generalist (or generic) foundation of social work practice has been inclusively built on system theory and the ecological system (ecosystem) perspective (see [50]), which, according to [26], are applicable at the micro-, mezzo-, and macro-practice levels for social change and to sustain society.

### 4.2. Policy Recommendations

The best practices of the performance economy (model)—services or commodities through rents or lease—and/or CE models (or a mixture of linear, performance, and CE) [51] is a good alternative to the popular traditional linear economy, which makes, uses, and disposes of waste materials as far as how solid wastes are generated and accumulated. Thus, any of these approaches could be anchored by social work in the best interest of business organizations, governmental levels, and every population group in Nigeria if the profession is given adequate support and cooperation with the relevant resources and necessary legislative framework.

The multicultural situation in Nigeria calls for a broader and culturally sensitive research evaluation and implementation in relation to SWM strategies, which can also create opportunities for simultaneous evaluation vis-à-vis good governance and the effects of policies and legislation on citizens. Social inclusion, citizen engagement, and knowledge creation are currently three of the eight guiding objectives of [52] SWM projects. Generally, current waste management practices in Nigeria must be understandable from the legal and policy instruments that are put in place and which call for constant evaluative research.

### 4.3. Limitations

Notwithstanding the recommendations, it seems appropriate to acknowledge the limitations of this RER. The number of databases that were searched did not seem to have freely available and valuable literature with relevant evidence of concern. For example, the literature such as [53] that implemented meta-analysis and meta-synthesis to examine theoretical key cultural factors and social practices influencing solid waste governance and management (SWG&M) in Nigeria was not available in those four databases via UEF Primo. In addition, [54] presented a systematic review of the literature on Nigeria’s WM in regard to waste characterization, waste management practices, ecological impacts, public–private partnership, ethical issues, and legal frameworks and [55] conducted a comprehensive review of the literature and critically evaluated the worsening conditions in municipal SWM in developing nations with a focus on Nigeria. Because the scholarly studies were not generated from this study’s searched databases, we may conclude that other studies with relevant evidence likely exist somewhere else. Nevertheless, the number of studies with relevant evidence of focus still seems very small, pro rata. The freely available scholarly studies beyond the reach of this RER support the fact that there is a huge gap between research and evidence-based policy implementation and practice in Nigeria.

## 5. Conclusions

This rapid review searched, examined, highlighted, and summarized or evaluated the best available or existing evidence for scholarship impact on sociocultural factors in Nigeria’s SWM strategies. Generally, the constant factors in the very few studies that revealed substantive evidence of sociocultural dimensions in Nigeria’s SWM practices in the literature seem to expose the inadequate and weak multidisciplinary or non-holistic approach to driving innovation and effective social impact in Nigeria’s WM practices. In other words, for more effective and efficient refuse management practices in Nigeria, there is an urgent need to examine the positions of social science interventions in existing SWM policies, laws, and practices. There is also a need to reconcile SWM policy documents and relevant evidence-based vis-à-vis holistic approach research because the emerging gaps are clearly due to an insufficient conceptual and theoretical basis for circular economy and waste management, on the basis of which it would be possible to build comprehensive legislation, professional expertise, policy strategies, and, ultimately, the necessary national and local waste management infrastructure, not forgetting the key role of citizens as grassroots waste managers.

## Figures and Tables

**Figure 1 ijerph-20-06245-f001:**
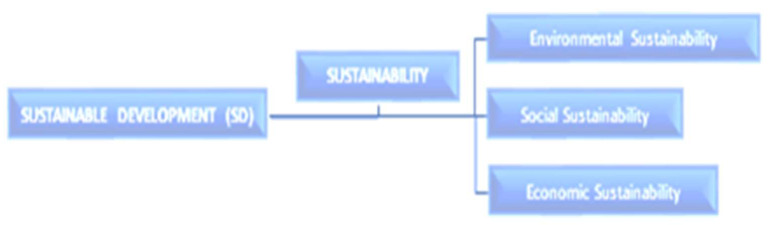
The hierarchy of the relationships among the main concepts of SD since 1987.

**Figure 2 ijerph-20-06245-f002:**
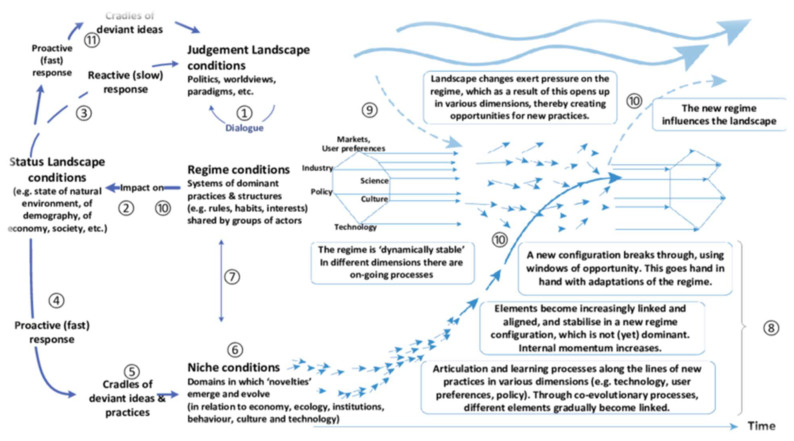
An adapted form of Wigboldus et al.’s (2001) multi-level perspective [42].

**Figure 3 ijerph-20-06245-f003:**
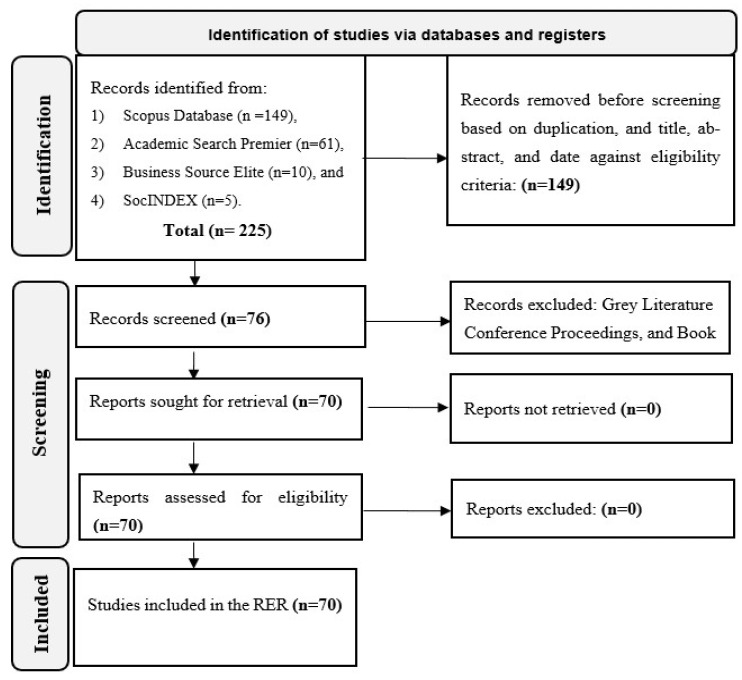
PRISMA 2020 flow diagram of the study selection process **[47]**.

**Figure 4 ijerph-20-06245-f004:**
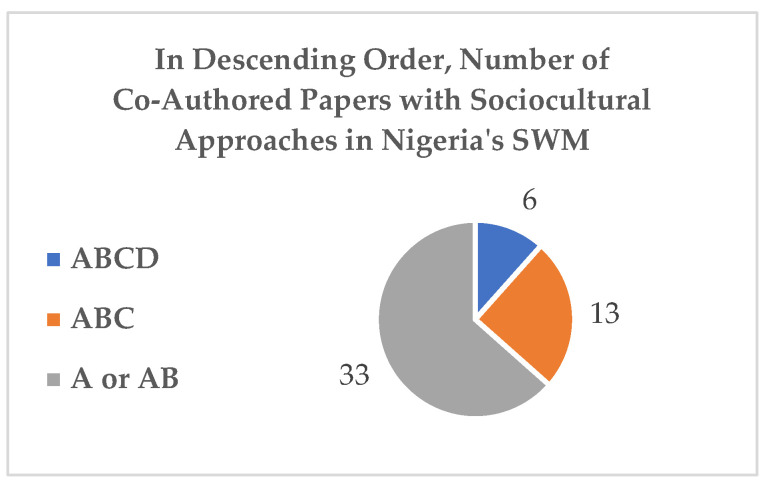
Group 1.

**Figure 5 ijerph-20-06245-f005:**
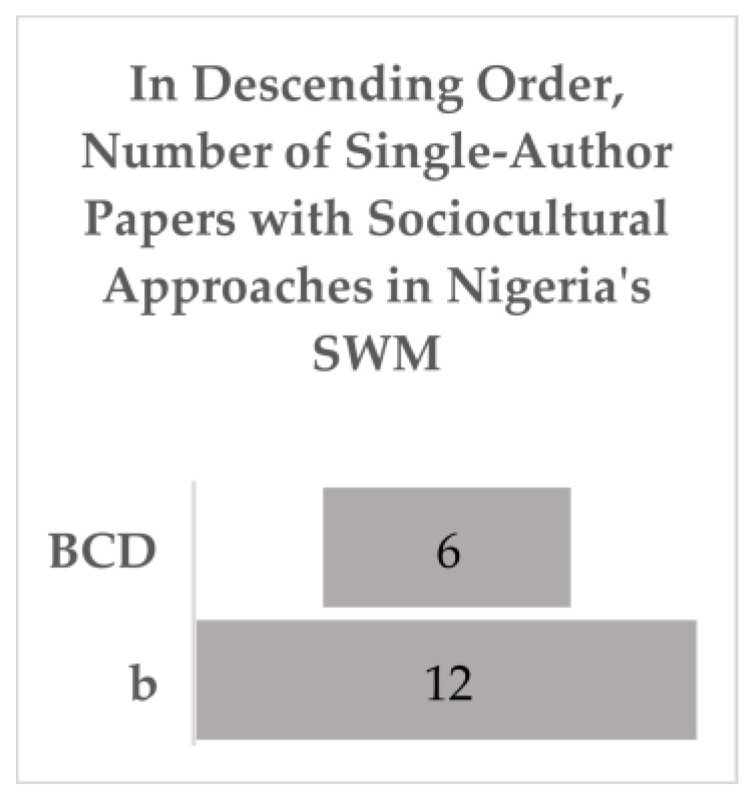
Group 2.

**Table 1 ijerph-20-06245-t001:** Abstracting the evidence, references, and frequencies.

Group (G)	Abstracting the Evidence (AE)	References
G1: Co-Authors	1. “…important to look for technology-based solutions to improve recycling/recovery activities …**active participation of the population**… attract and unite people to achieve such common goals…the importance of **increasing awareness-raising** to **change public perception** towards waste from being a nuisance to a valuable resource.” “…**attitudinal factors**…” “…uphold…**knowledge** and **values**.” “…**advocacy**…**social marketing**.”	SP2, SP9, SP7, SP10, SP13, SP16, SP19, SP32, SP40, SP46, SP47, SP48, SP50.
	2. “…**involvement of the plurality of the non-state actors** in waste management **co-production** …” “…**dangers of poor sanitary habits**…”	SP2, SP8. SP13, SP17, SP50, SP52.
	3. “… the **adoption of a holistic** and **indigenous approach**…”	SP7, SP21, SP38,
	4. “… these **laws** are inadequate to meet the contemporary challenges…”	SP13, SP19, SP35, SP40.
G2: Single Authors	5. “A major limitation to the integration of informal waste collectors and scavengers is the **social acceptance** of their activity as a viable source of income, and of themselves as environmental agents in the sustainability of virgin resources.”	SP27.
	6. “The study recommends inter alia **that practitioners should be cognizant of the use of waste management plans** (WMPs) to minimize all categories of waste, especially materials, on their projects.”	SP34.
	7. “Findings show that the Policy Implementors preferred the use of personal contact as the channel for **disseminating environmental information**, whereas the Policy Formulators relied on the use of posters, radio/TV talks, and professional meetings. Some barriers to disseminating information to the public included: lack of access to information sources, lack of standards for acquisition of information, and lack of funds to publish information materials.”	SP36.
	8. “A **public enlightenment campaign** is recommended for proper **education of the public** on modern ways of garbage disposal.”	SP41.
	9. “Successful solid waste management in Nigeria will require a **holistic program** that will integrate all the technical, **economic**, **social**, **cultural**, and **psychological factors** that are often ignored in solid waste programs.”	SP42.
	10. “It highlights the roles of **individuals**, **households**, changing lifestyles and diet, business cycles, residential segregation, and state and **non-state actors** and institutions in waste management in Lagos.”	SP45.

NB: The bolded (words, phrases, and texts) are aspects of the social science/social work approach to SWM.

**Table 2 ijerph-20-06245-t002:** Frequency and intensity effect sizes.

	SP02 [Ezeudu et al., 2021]	SP07 [Egun and Evbayiro, 2020]	SP08 [Nzeadibe and Ejike-Alieji., 2020]	SP09 [Oh and Hettiarachchi, 2020]	SP10 [Oyije et al., 2020]	SP13 [Olukanni and Nwafor, 2019]	SP16 [Amusan et al., 2018]	SP17 [Ike et al., 2018]	SP19 [Baaki et al., 2017]	SP21 [Mbah and Nzeadibe, 2017]	SP27 [Oguntoyibo 2012]	SP32 [Nzeadibe and Ajaero, 2011]	SP34 [Oladiran O. J., 2009]	SP35 [Adedeji and Ako, 2009]	SP36 [Solomon, 2009]	SP38 [Onwurah et al., 2006]	SP40 [Ogbonna et al., 2002]	SP41 [Ezeronye, 2000]	SP42 [Agunwamba, 1998]	SP45 [Olukoju 2019]	SP46 [Oladiti et al., 2019]	SP47 [Iheanacho et al., 2018]	SP49 [Babayemi et al., 2017]	SP50 [Ogbonna and Mikailu, 2019]	SP52 [Mbah and Nzeadibe, 2017]	Frequency
A1	1	1		4	1	1	2		1			2					3				3	1	3	1		13/25
A2	2		1			1		2																	1	5/25
A3		1								1						1								1		4/25
A4						1			1					1			1									4/25
A5											1															1/25
A6													1													1/25
A7															1											1/25
A8																		2								1/25
A9																			4							1/25
A10																				3						1/25
**Intensity**	**3/39**	**2/39**	**1/39**	**4/39**	**1/39**	**3/39**	**2/39**	**2/39**	**2/39**	**1/39**	**1/12**	**2/39**	**1/12**	**1/39**	**1/12**	**1/39**	**4/39**	**2/12**	**4/12**	**3/12**	**3/39**	**1/39**	**3/39**	**2/39**	**1/39**	

## Data Availability

Data will be made freely available upon request.

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
