# Peer review of "A Rapid Review of Sociocultural Dimensions in Nigeria’s Solid Waste Management Approach"

_ijerph, 2023, doi:10.3390/ijerph20136245_

Round 1
Reviewer 1 Report
In this manuscript, the authors conducted a rapid evidence review of likely interdisciplinarity and/or transdisciplinary and multidisciplinary approaches in the literature on Nigeria’s SWM system. The manuscript is well written and organized. However, major issues to be addressed were recognized. Therefore, following comments are included to revise this manuscript. Authors are encouraged to resubmit the manuscript after addressing following comments.
1. What progress has been made in this work? new method?
2. Why it is necessary to conduct a rapid evidence review of likely interdisciplinarity and/or transdisciplinary and multidisciplinary approaches in the literature on Nigeria’s SWM system.
3. Papers are mostly a list of literature and lack critical comments.
4. This article is a rapid review. Is title ”2. Materials and Methods” appropriate?
5. The titles of Figures 4 and 5 are not appropriate, and it is recommended to modify them.
6. As a recommendation, the picture is too simple and not clear.
7. Table 2 layout problem, cannot see the contents of the proposed changes.
The writing language and logic of the paper need to be further polished.
Author Response
Dear Reviewer,
Thank you for your comments! They are appreciated.
- The rapid evidence review is not about methods, and the Keywords section does not include methods either. But the Title has been further modified to reflect more the focus of the study. The focus is a search for evidence of sociocultural dimension in scholarly work focusing on Nigeria’s SWM. And the study found inadequate substantive evidence in this regard and made appropriate recommendations. In this way, the conceptual and theoretical bases for Nigeria’s SWM approaches seems the reason for the persistent waste disposal problems in the country. Therefore, the paper argued for the interdisciplinarity, and/or the transdisciplinary and multidisciplinary approaches needed to build a holistic theoretical basis for SWM strategies in Nigeria.
- In addition, because of the multiple disciplines’ trends in the issue of SWM generally, the RER was not designed for a critical analysis but mainly about search for evidence of sociocultural dimensions in scholarly literature focusing on Nigeria’s SWM.
- Lines 52 – 54 (formerly 46 – 47) stated the reasons for adopting a Rapid Evidence Review (RER).
- The title of the Method section has been edited in the manuscript as appropriate, and the irrelevant words deleted.
- Similarly, the Tables and Figures in the manuscript have been modified and edited in response to your observations. The problems with the Tables and Figures were partly caused by the journals template, and the journal has been mandated to help make necessary adjustment during production.
- Lastly, because the paper was initially edited by Scribendi.com, we decided to make some changes as a response to you comment. But the journal may be mandated to further edit the language. Thank you!
Reviewer 2 Report
Dear authors,
thank you very much for your manuscript, which deals with a very interesting topic related to sociocaltural dimensions of environmental management.
However, the manuscript must be considerably improved in order to be acceptable for publication.
The 238 out of 441 lines of the text have been used for presenting introductory and background information. The first impression of the reader is that the same issues are discussed again and again, while significant information about the solid waste management system of Nigeria is missing. The only quantitative data is this of country's population and SW quantities per capita and day. To understand the economic, environmental, and sociocultural framework of SW management, the reader needs additional information about the national laws and regulations, infrastructures, applied disposal methods, recycling programs, dangerous and special waste management, waste imports, good practices and failures, etc. Moreover, since SW management policies and practices are a political issue as well, it is necessary to discuss the changes that have been made in the last decades and your criticism (inadequate, non-holistic approach) must be based on facts and relevant references.
Please, avoid the imperative form to introduce the references (see...).
The quality of Figures 1,2 and 5 is very poor. Actually, figure 1 does not contains more information than the text.
The titles of figure 4 and 5 must describe in detail what is presented.
In Table 1 there is no need to describe the abstractive evidences using phrases from the revised papers.
In Table 2 it is not clear the meaning of the first and last column (the last column is not the sum of the previous ones?). Also, the format of the table must be improved (spacing in first raw, width of last column).
In the section of the results, you must present first the number of papers that are A and non-A, B and non-B, etc., and then to search for combinations BCD, ABCD, etc.
Line 381: in agreement with 26 and 26...
Author Response
Dear Reviewer,
Thank you for your comments!
- It is necessary to state the reason for adopting a Rapid Evidence Review (RER) approach instead of Systematic Literature Review (SLR) in the background of the paper. And the reasons are now in lines 52 – 54 with explanation of the RER, techniques of analysis and reporting of findings. In addition, we want to emphasize that because of the multiple disciplines’ trends in the issue of SWM, the RER was not designed for a critical analysis but mainly about search for evidence of sociocultural dimensions in scholarly literature focusing on Nigeria’s SWM. Thus, we argued for the interdisciplinarity, and/or the transdisciplinary and multidisciplinary approaches needed to build a holistic theoretical basis for SWM strategies.
- Similarly, the approach to introducing the references and line 381 (now line 394) had been edited in the manuscript. But the complex information search and analysis recommended is far beyond the scope of this study. As a matter of fact, a document analysis study of those subject areas suggested is ongoing but as a separate manuscript that will be submitted for publication very soon. The authors are also sensitive to possible political dimension of studying SWM. Nevertheless, our goal is to join forces to strengthen existing practices. In addition, the analysis technique recommended does not conform to Qualitative Meta-summary method adopted in the study. Table 2 (now Figure 6) supports Table 1, and this is the essence of the method for clarity and replication purpose. In particular, the Boolean Algorithm ABCD or abcd is for only comparative grouping of the characteristics of evidence found in the review. And the comparative groupings also support Table1 and Table 2 (now Figure 6) as the practice in Qualitative Meta-summary technique. Graphs have not been helpful in the reporting.
- Furthermore, the Tables and Figures in the manuscript have been modified and edited in response to your observations. The problems with the Tables and Figures were partly caused by the journals template, and the journal has been mandated to help make necessary adjustment during production.
- Lastly, Line 394 (formerly line 381) had been edited to read …in agreement with 26 and 27.
Thank you so much!
Reviewer 3 Report
Reviewers' comments:
Authors should pay attention to the revisions, addressing each and every comment.

Author Response
Dear Reviewer
- Thank you so much for your comments! The Title has been further modified to reflect more the focus of the study.
- Sorry! The Abstract was modified according to the requirement of the journal. In other words, that is what the template mandated us to do with the abstract.
- Keywords have been edited accordingly.
- Tables and Figures in the manuscript are now modified and edited in response to your observations. The problems with the Tables and Figures were partly caused by the journals template, and the journal has been mandated to help make necessary adjustment during production.
- Please, your comments (no 6) are not clear to me! But the gaps in the tables are the effects of the journal templates on the manuscript. But additional adjustments have been implemented to make the tables clearer. Table 1 has references but coded as SP1, SP2, etc., as well as Figure 6 (formerly Table 2).
- Appendix A (now A1) is according to the structure of the journals template. However, the journal is expected to assist in fine tuning the Tables and Figures during production.
- In the Conclusions section, we avoid being unnecessarily repetitive. Lines 444- 446 are about the aim of the study. While lines 446 - 449 is about the results, lines 450 - 454 are about the next levels to strengthen SWM in the country. In essence, your comments are appreciated, but these are the most important elements in any conclusions to a journal manuscript, particularly a manuscript of this nature where repetition is to be avoided. As a matter fact, one of the reviewers cautioned about being repetitive in the manuscript.
- Lastly, the DOIs, where available, have also updated.
Your comments are appreciated!
Round 2
Reviewer 2 Report
Dear authors,
unfortunately, I don't see any significant improvements in the current version of your article over the previous one. The lack of literature on the sociocultural impacts of waste management, the absence of multidisciplinary planning and the delays in adopting circular economy principles characterize the waste management practices applied in 90% of the countries in the world, including my country. Thus, conclusions of this type neither contribute to the scientific knowledge nor make the article interesting and informative for the reader.
I apologize for not being able to be pleasant with my review.
Author Response
Dear Reviewer,
Please, accept our huge thanks for your comments! Your feelings have been further taken into consideration in the manuscript. We have tried to emphasize the ubiquitousness of SWM in many countries and not as a problem limited to Nigeria alone, and in response to the connotation in your statistic (90%). In other words, some new significant changes have been implemented in the manuscript in response to your feelings and views.
Nigeria is not a strange place, and the hazards of solid waste in the country are in the limelight in empirical research papers across the world. So, as iterated before, our aim is to create an evidence-based platform to holistically strengthen existing SWM across many countries, including Nigeria. Of course, existing interventions in such countries are also taken into account because of the tendency to politicize the issue at stake.
Once more, thank you for your efforts.